# Transmission dynamics and control of multidrug-resistant *Klebsiella pneumoniae* in neonates in a developing country

Thomas Crellen[1,2]*, Paul Turner[1,2,3], Sreymom Pol[1,3], Stephen Baker[2,4], To Nguyen Thi Nguyen[4], Nicole Stoesser[2], Nicholas PJ Day[1,2], Claudia Turner[1,2,3], Ben S Cooper[1,2]*

[1]Mahidol-Oxford Tropical Medicine Research Unit, Faculty of Tropical Medicine, Mahidol University, Bangkok, Thailand; [2]Nuffield Department of Medicine, University of Oxford, Oxford, United Kingdom; [3]Cambodia-Oxford Medical Research Unit, Angkor Hospital for Children, Siem Reap, Cambodia; [4]Oxford University Clinical Research Unit, Centre for Tropical Medicine, Ho Chi Minh City, Viet Nam

**Abstract** Multidrug-resistant *Klebsiella pneumoniae* is an increasing cause of infant mortality in developing countries. We aimed to develop a quantitative understanding of the drivers of this epidemic by estimating the effects of antibiotics on nosocomial transmission risk, comparing competing hypotheses about mechanisms of spread, and quantifying the impact of potential interventions. Using a sequence of dynamic models, we analysed data from a one-year prospective carriage study in a Cambodian neonatal intensive care unit with hyperendemic third-generation cephalosporin-resistant *K. pneumoniae*. All widely-used antibiotics except imipenem were associated with an increased daily acquisition risk, with an odds ratio for the most common combination (ampicillin + gentamicin) of 1.96 (95% CrI 1.18, 3.36). Models incorporating genomic data found that colonisation pressure was associated with a higher transmission risk, indicated sequence type heterogeneity in transmissibility, and showed that within-ward transmission was insufficient to maintain endemicity. Simulations indicated that increasing the nurse-patient ratio could be an effective intervention.

**\*For correspondence:**
thomas.crellen@ndm.ox.ac.uk (TC);
Ben@tropmedres.ac (BSC)

**Competing interests:** The authors declare that no competing interests exist.

## Introduction

Infections with multidrug-resistant Enterobacteriaceae constitute a major threat to public health in all regions of the world (*Schwaber and Carmeli, 2008*; *Theuretzbacher, 2017*) and extended spectrum β-lactamase (ESBL) producing and carbapenem-resistant Enterobacteriaceae have been prioritised by the World Health Organization as pathogenic bacteria in need of novel therapeutics (*World Health Organization, 2014*; *World Health Organization, 2017*). These organisms pose the highest risk to subgroups of patients such as those undergoing surgery, requiring invasive devices, and neonates (*Peleg and Hooper, 2010*; *Goldmann, 1981*). Antimicrobial resistance is of particular concern in developing (lower and middle-income) countries where the estimated per capita mortality from drug-resistant bacteraemia is far greater than in high-income countries and where last-line antibiotics may be unavailable or unaffordable (*Lim et al., 2016*; *Zaidi et al., 2005*). The high risk of difficult-to-treat nosocomial infections threatens to undermine patient confidence in developing world hospitals and health systems (*Dondorp et al., 2018*).

In this paper, we focus on *Klebsiella pneumoniae*, which is amongst the most clinically important multidrug-resistant Gram-negative bacteria in developing country settings (*Fox-Lewis et al., 2018*; *Zellweger et al., 2017*; *Musicha et al., 2017*). Genomic studies characterising the population

structure of *K. pneumoniae* have revealed a complex consisting of three separate species (*K. pneumoniae*, *K. quasipneumoniae*, *K. variicola*) that are indistinguishable by culture or standard biochemical assays (*Holt et al., 2015*). Where these isolates remain undifferentiated by molecular assay, we refer to them as *K. pneumoniae sensu lato* (in the broad sense).

While there is currently widespread concern about Gram-negative bacteria as an emerging threat due to high levels of plasmid-borne resistance (*Nordmann et al., 2011*), pathogens such as *K. pneumoniae* have been considered a major problem in nosocomial settings for over half a century (*Yow, 1955*). Research during the 1960s identified a number of drivers of *Klebsiella* colonisation and infection in hospital settings including invasive devices (*Mertz et al., 1967*), environmental contamination (*Kresky, 1964*), introduction from the community (*Kessner and Lepper, 1967*), person-to-person transmission (*Weil et al., 1966*), endogenous selection from antibiotic pressure (*Selden et al., 1971*) and transient carriage on the hands of health-care workers (*Adler et al., 1970*). In a comprehensive review of this body of work, Montgomerie concluded that 'The likely means of transmission of *Klebsiella* is via the hands of hospital staff members' (*Montgomerie, 1979*).

More recently, carriage studies in high-income hospital settings in temperate regions have used whole-genome sequencing to show the critical importance of asymptomatic carriage for understanding the epidemiology of *K. pneumoniae*, establishing a firm link between gastrointestinal carriage and clinical infection (*Martin et al., 2016*; *Gorrie et al., 2017*; *Gorrie et al., 2018*). In contrast to studies in high-income countries where multidrug-resistant *K. pneumoniae* is typically rare, prospective carriage studies in hospitalised paediatric populations in developing countries in Africa and Asia have reported hyperendemic levels of ESBL-producing Enterobacteriaceae including *K. pneumoniae* (*Andriatahina et al., 2010*; *Roberts et al., 2019*; *Founou et al., 2019*; *Turner et al., 2016*) consistent with the greater burden of disease due to these organisms in lower income settings (*Lim et al., 2016*; *Musicha et al., 2017*).

Despite their clinical importance, there are major gaps in our knowledge of the epidemiology of multidrug-resistant *K. pneumoniae*. First, while a number of recent investigations of localised hospital outbreaks in high-income settings have provided evidence that long-term environmental contamination of sinks and other sites may play a role (*Clarivet et al., 2016*; *Decraene et al., 2018*; *Mathers et al., 2018*), the relative importance of persistently contaminated point sources versus patient-to-patient transmission in endemic settings remains unclear. Second, though it is widely assumed that antibiotic exposures play an important role in selecting for multidrug-resistant *K. pneumoniae* (*Baker et al., 2018*), such effects have not previously been quantified at the patient level in a way that disentangles antibiotic effects from general exposure to the hospital environment. Third, attempts to quantify transmissibility and how this varies by sequence type (ST) are lacking. Fourth, the impacts of other factors that might plausibly affect transmission including staffing levels, infant breast feeding and use of probiotics have not been explored.

To address these knowledge gaps, we used data collected from a year-long prospective observational carriage study in a Cambodian neonatal intensive care unit and analysis methods which build upon a previously described data-driven stochastic model (*Forrester and Pettitt, 2005*). We fit four models with logit link functions to estimate the impact of covariates on the daily risk of acquisition of 3GC-R *K. pneumoniae s.l.*. As these models are unable to identify the force of infection, and genomic data show the ward 3GC-R *K. pneumoniae s.l.* to be a highly heterogeneous bacterial community, we then fit five linear transmission models to estimate the force of infection for acquisition of each ST. Finally, we use the estimated parameters to provide model-based assessments of the potential impact of hypothetical control measures. We define "acquisition" as the first detection of the organism within an infant following an initial negative swab on admission, which may indicate transmission from other colonised infants or hospital staff, or endogenous selection as a result of antibiotic pressure.

## Results

### Descriptive epidemiological data

Over the year-long observation period, there were consistently high rates of patient carriage of third generation cephalosporin-resistant (3GC-R) *K. pneumoniae sensu lato*. Of 333 infants admitted to the neonatal unit, 121 of 289 (42%) were found to be colonised on the first swab taken within 48

hours of ward admission. A further 21 out of 44 (48%) were positive on the first swab that was taken more than 48 hours after admission. Overall, 109/191 (57%) infants who initially screened negative for 3GC-R *K. pneumoniae s.l.* became positive during their stay in the neonatal unit. Almost all 3GC-R *K. pneumoniae s.l.* isolates were ESBL producers (1412/1423; 99%), and only 5/1423 (0.35%) were resistant to imipenem. Co-colonisation with 3GC-R *E. coli* was observed in 52 infants on their first swab, and a further 102 infants became co-colonised with both resistant organisms during their stay on the neonatal unit. Full details on the study population, including blood-stream infections and mortality have been reported previously (*Turner et al., 2016*), and a summary is provided in *Table 1*.

The daily counts of infants known to be colonised with 3GC-R *K. pneumoniae s.l.* (*Figure 1A*), shows no clear trend but large stochastic fluctuations, which are expected given the small size of the ward (eight beds) and frequent discharges of patients and introduction of colonised infants (imported carriage). A representation of swabbing interval outcomes, as used in the models, is shown in *Figure 1B*. While the median length of stay was four days, the distribution is highly skewed with a tail of long-staying patients (*Figure 1C*). The most frequently used antibiotic combination was ampicillin with gentamicin which was used empirically to treat suspected sepsis in infants admitted from the community and was taken on one fifth (19%) of patient days on the ward. This was followed by imipenem, which was taken on 12% of patient days. Imipenem was typically used when culture results showed non-susceptibility to first-line antibiotic choices and empirically in infants with suspected hospital-acquired infection. All other antibiotic combinations were used at a much lower frequency (*Figure 1D*).

## Factors associated with carriage acquisition

Infants were prospectively screened for the organism whilst on the ward by culture of rectal swabs on selective media. For infants with negative cultures for 3GC-R *K. pneumoniae s.l.* on admission to the ward, the outcome (acquisition of the organism) was therefore interval censored between subsequent rectal swabs that were taken a median of 2 days apart (IQR 1, 3 days). In total there were 400

**Table 1.** Summary of characteristics of infants admitted to the neonatal intensive care unit at a children's hospital in Cambodia from September 2013 to September 2014.
Colonisation status with third generation cephalosporin-resistant *Klebsiella pneumoniae sensu lato* was recorded through prospectively taken rectal swabs.

| Variable | Males | Females | Total |
|---|---|---|---|
| Number of Patients | 177 (53.1%) | 156 (47.8%) | 333 (100%) |
| Length of Stay in Days Median, (IQR)[*] | **6 (4, 11) | 6 (4, 11) | 6 (4, 11) |
| Colonised with *K. pneumoniae* at Entry (or Unknown Time)[†] | 66 (10) | 55 (11) | 121 (21) |
| Colonised with K. pneumoniae During Admission | 54/101 (54.5%) | 55/90 (61.1%) | 109/191 (57.1%) |
| Co-colonised with *K. pneumoniae* and *E. coli* at Entry (or Unknown Time)[†] | 26 (5) | 19 (2) | 45 (7) |
| Co-colonised with *K. pneumoniae* and *E. coli* During Admission | 49/146 (33.6%) | 53/135 (39.3%) | 102/281 (36.3%) |
| Age at Entry in Days (IQR)[*] | *8 (2, 15) | 9 (1, 17) | 8 (1, 16) |
| Probiotic Taken[‡] | 76/177 (42.9%) | 62/156 (39.7%) | 138/333 (41.4) |
| Breast Milk Fed | 163/177 (92.1%) | 139/156 (89.1%) | 302/333 (90.1%) |
| Severe[§] | 35/177 (19.8%) | 32/156 (20.5%) | 67/333 (20.1%) |
| Born Premature | 30/177 (16.9%) | 24/156 (15.4%) | 54/333 (16.2%) |

[*] Interquartile range. [†] Colonised at entry is defined as an initial positive swab within the first 48 hours of admission; if the first swab is positive and it was taken later than 48 hours from admission then the infant is considered to be colonised at an unknown time. [‡] Assigned by clinician to receive oral *Lactobacillus acidophilus.* [§] Severe symptoms are requiring ventilation, continuous airway pressure or inotopes.

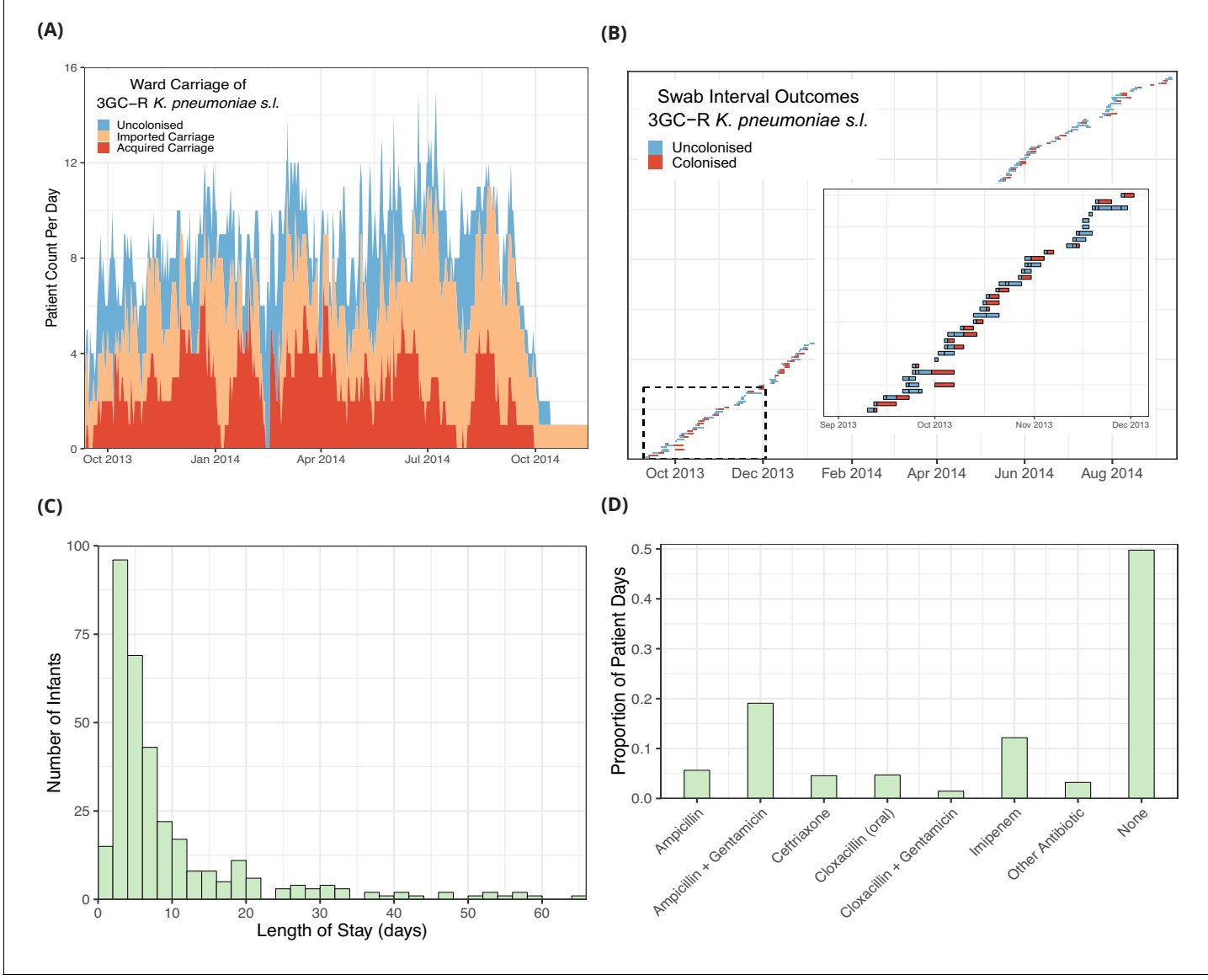

**Figure 1.** Descriptive epidemiological data from a cohort of 333 infants admitted to a neonatal unit in a Children's Hospital in Cambodia from September 2013 to September 2014. Daily counts of neonates colonised with third generation cephalosporin-resistant (3GC-R) *Klebsiella pneumoniae sensu lato* over the study period are shown in panel A, where colour reflects uncolonised, imported or acquired cases, according to case definitions. The total height of the peaks shows the ward occupancy on that day. The results from rectal swabs among the 191 infants uncolonised at entry for 3GC-R *K. pneumoniae s.l.* are shown in panel B, with the window highlighting the swab outcomes from the first thirty five infants uncolonised at entry. Each row represents a patient and each coloured block represents a swab interval, where the width is the number of days in the interval (i.e. time between swabs). Outcomes are shown up to the first swab positive for 3GC-R *K. pneumoniae s.l.*, after which time the patient is assumed to be colonised until discharge. The length of stay distribution for infants in the neonatal unit is shown as a histogram in panel C, where the bin width is two days. An infant's length of stay is the total time in the neonatal unit during the study period, including re-admissions. The 333 infants were present in the neonatal unit for a total of 3417 study days. The proportion of study days when infants took the six most common antibiotic combinations, or other antibiotics, or none are shown in panel D.

swab outcomes (either negatives or a first positive swab) over 864 patient days from 191 infants with a negative culture at entry.

Four models were fitted to the interval censored swab data to determine factors associated with daily risk of carriage acquisition. The best performing single intercept model (model A; *Table 2*) with the lowest WAIC (see Methods) considered exposure to antibiotics in the previous 96 hours and did not include a term for colonisation pressure (i.e. daily per-patient acquisition risk did not depend on

**Table 2.** Comparison of models for the risk of acquiring third generation cephalosporin-resistant *Klebsiella pneumoniae sensu lato* over 864 patient days in a neonatal intensive care unit in Cambodia.

Models vary by explanatory variables (A-C) or by permitting the intercept to vary between study months in a hierarchical model (D). Models were fitted on the log-odds scale with a logit link function, hence prior distributions are shown as log-odds. Posterior parameter distributions have been transformed using the logistic function and are shown as probabilities. Prior distributions are normal distributions, shown in brackets are the mean and standard deviation respectively.

| Risk Factor Model | Parameters | Priors | Posterior Median (95% CrI)[*] | WAIC[†] |
|---|---|---|---|---|
| (A) Single intercept Standard covariates[‡] 96 hour antibiotic exposure | α (intercept) β (slopes) | normal(0, 10) normal(0, 5) | 0.23 (0.055, 0.60) ORs[§] in results | 438 |
| (B) Single intercept Standard covariates[‡] 48 hour antibiotic exposure | α (intercept) β (slopes) | normal(0, 10) normal(0, 5) | 0.26 (0.068, 0.63) Not shown | 441 |
| (C) Single intercept Standard covariates[‡] + colonisation pressure term[¶] 96 hour antibiotic exposure | α (intercept) β (slopes) | normal(0, 10) normal(0, 5) | 0.26 (0.059, 0.64) Not shown | 440 |
| (D) Intercept varies by month Standard covariates[‡] 96 hour antibiotic exposure | α[month] (intercept) μ (normal mean) σ (normal standard deviation) β (slopes) | normal(μ, σ) normal(0, 3) half-normal(0, 1) normal(0, 3) | Varies by month[††] 0.21 (0.044, 0.57) 0.54 (0.51, 0.63) Not shown | 440 |

[*] 95% Credible interval. [†] Widely applicable information criterion (a model comparison statistic where lower values indicate better fitting models). [‡] Standard covariates: use of ampicillin, ampicillin + gentamicin, cloxacillin (oral), ceftriaxone, cloxacillin + gentamicin, and imipenem within the previous 48 or 96 hours; whether breast fed; receipt of an oral probiotic on entry (*Lactobacillus acidophilus*), sex, premature (born before the 37th week of pregnancy), severity (defined as severe if requiring ventilation, continuous positive airway pressure or inotropes), already colonised with 3GC-R *E. coli*, age in days on first admission to the NU, and the daily number of nurses on the ward. These explanatory variables were treated as binary and, where appropriate, time-varying. Covariates were recorded for every day the infant was present in the neonatal unit (see Methods for full details). [§] Odds ratios. [¶] Colonisation pressure is the number of known colonised patients on the ward on a given day. [††] Median posterior probability ranges by month 0.20–0.23.

the number of other patients who were colonised on a given day). Models with a 48 hour antibiotic exposure period (model B) and those that included a colonisation pressure term (model C) showed slightly worse fits.

The covariates associated with reduced daily risk of acquisition were breast milk feeding (odds ratio [OR] 0.69 [95% CrI 0.35, 1.41]) and increasing the number of nurses, for instance three nurses in the ward was associated with an OR of 0.55 (95% CrI 0.15, 1.77) relative to zero nurses (baseline). Male sex was also associated with a reduced risk of acquisition (OR 0.68 [95% CrI 0.43, 1.04]). Other covariates such as taking a probiotic (OR 0.88 [95% CrI 0.55, 1.41]), a severe condition (OR 1.10 [95% CrI 0.55, 2.13]), prior colonisation with 3GC-R *E. coli* (OR 1.07, [95% CrI 0.65, 1.75]), and age at admission (10 days compared with zero days, OR 0.95 [95% CI 0.73, 1.24]) had ORs distributions centred closer to unity. As anticipated (though, to our knowledge, not previously shown), antibiotics taken (intravenously, with the exception of cloxacillin) within the past 96 hours were mostly associated with an increased risk of colonisation with 3GC-R *K. pneumoniae s.l.*. Ampicillin (OR 1.77 [95% CrI 0.88, 3.26]), ampicillin with gentamicin (OR 1.96 [95% CrI 1.18, 3.36]), ceftriaxone (OR 1.85 [95% CrI 0.68, 4.54]), oral cloxacillin (OR 1.49 95% CrI 0.47, 4.02]), and cloxacillin with gentamicin (OR 1.94 [95% CrI 0.55, 5.66]) were all associated with an increased risk of acquisition. Only intravenous imipenem (OR 1.01 [0.40, 2.30]) had a posterior distribution centred near unity, consistent with the carbapenem sensitivity of 3GC-R *K. pneumoniae* found in this setting. See **Figure 2A** for odds ratio posterior distributions.

Using the covariate posterior distributions, we also estimated the probability of colonisation for each of the 864 patient days where patients were at risk for acquiring 3GC-R *K. pneumoniae s.l.*. The median daily probability of first acquisition of 3GC-R *K. pneumoniae s.l.* for an infant was estimated from the best fitting model as 0.15. There is considerable variability in the risk of acquisition between patient days and the medians of the posterior probability distribution ranges nearly eight-fold from 0.047 to 0.35 (**Figure 2B**).

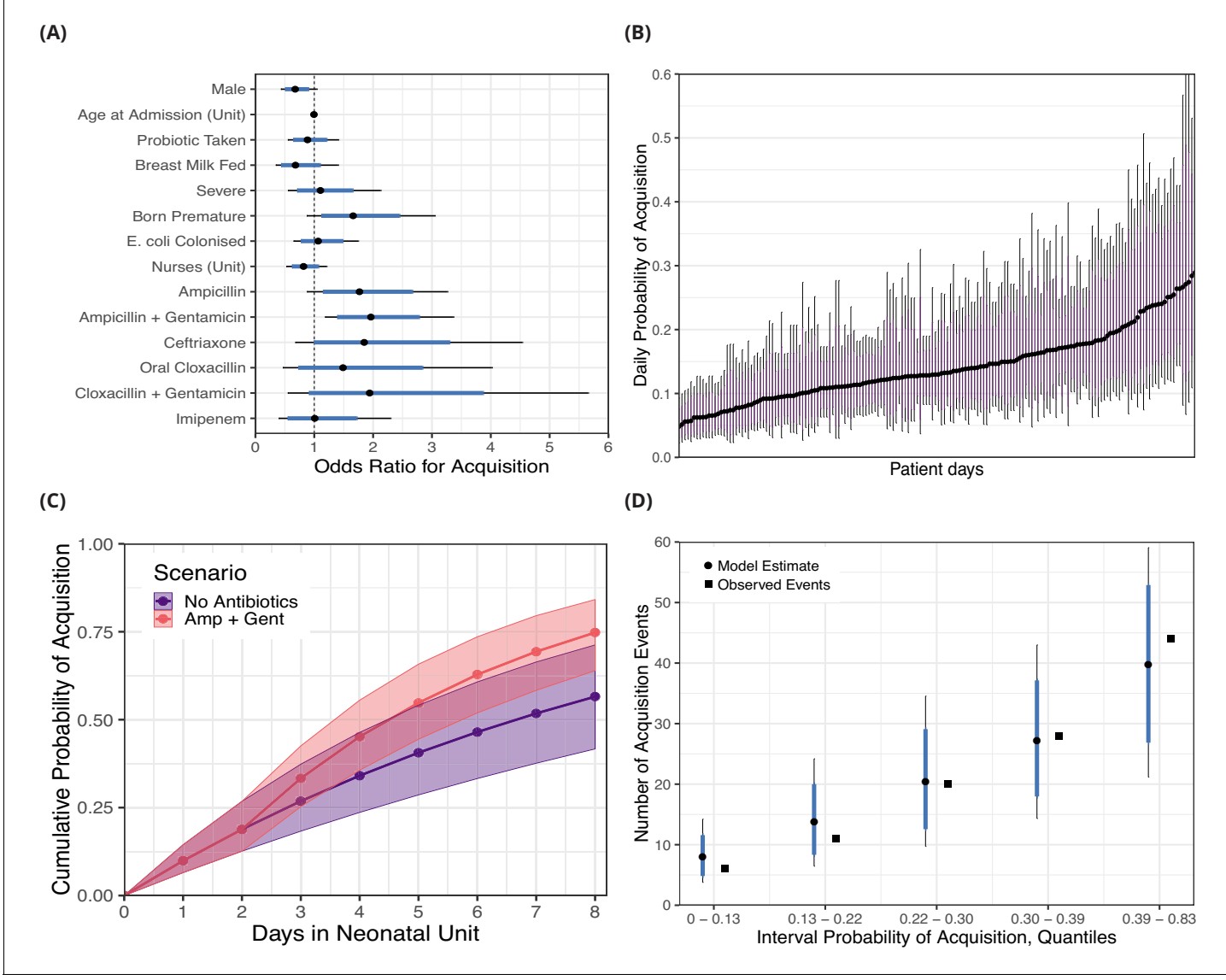

**Figure 2.** Posterior distributions for risk factors for the daily probability of acquiring third-generation cephalosporin-resistant (3GC-R) *Klebsiella pneumoniae sensu lato* among 191 susceptible neonates. Odds ratios for the daily risk of colonisation are shown in panel **A**. The daily risk of colonisation per patient day is shown in panel **B**. Note that the 864 patient days have been thinned by a factor of five for visualisation. The cumulative risk for different patient scenarios is explored in panel **C**; a four day old girl, born full term, without severe conditions, breast milk fed and not taking antibiotics or probiotics over eight days in the neonatal unit is shown in blue. The red line shows the same infant, however ampicillin + gentamicin is taken from day three onwards. The lines and points in both cases show the cumulative probability posterior median, and the shaded area shows the 80% credible interval (CrI). In panel **D**, we took the probability of colonisation for each of the 400 swab interval and binned them into five quantiles. We then compared the expected number of colonisation events predicted by the model with the observed number of colonisation events (squares) in the swab intervals by quantile. In panels A, B and D points represent posterior medians, thick blue/purple lines represents the 80% CrI and thinner black lines represent the 95% CrI.

The online version of this article includes the following figure supplement(s) for figure 2:

**Figure supplement 1.** Posterior chains from Hamiltonian Markov chain Monte Carlo fitting using Stan for risk factor model A (see *Table 2*).

**Figure supplement 2.** Estimates from risk factor model A with variable priors.

The risk of becoming colonised with 3GC-R *K. pneumoniae s.l.* is cumulative over an infant's length of stay and varies in response to interventions, such as consumption of antibiotics. We show the cumulative risk of first acquisition under two scenarios: 1) where an initially four day old, breastfed, female infant remains in the ward for eight days without taking antibiotics or probiotics;

and 2) where an infant with the same characteristics is prescribed ampicillin with gentamicin from day three onwards. The median cumulative risk of acquiring 3GC-R *K. pneumoniae s.l.* after eight days for the first scenario is 0.57 (80% CrI 0.42, 0.71) and for the second scenario is 0.75 (80% CrI 0.64, 0.85). Although the median cumulative risk between the two scenarios diverges the longer the infants are in the neonatal unit, the uncertainty also increases with time (*Figure 2C*).

## Swab sensitivity

We estimated the sensitivity of rectal swabs for detecting 3GC-R *K. pneumoniae s.l.* by examining the swabs that followed a positive from the same patient. There were 936 such swabs which were taken from a patient after at least one swab positive for 3GC-R *K. pneumoniae s.l.*, and 90 (9.6%) of these were negatives. Under the assumption that all negatives following a positive culture are false negatives, the false negative rate posterior median was 0.096 (95% CrI 0.078, 0.12) and the posterior median swab sensitivity was 0.90 (95% CrI 0.88, 0.92). Under the second assumption that three or more consecutive negative swabs following a positive culture represent a true decolonisation event, there were 72 false negatives, giving a false negative rate of 0.073 (95% CrI 0.058, 0.091) and a swab sensitivity of 0.93 (95% CrI 0.91, 0.94).

## Model assessment and comparison

The measures of Markov chain convergence showed high effective sample sizes (>400) and $\hat{R}$<1.01, indicating that the chains had run for long enough and had mixed well (see Methods and *Figure 2—figure supplement 1*). Model assessment was performed with a posterior predictive check; we estimated the probability of acquisition for each of the 400 swabbing intervals and binned these probabilities into groups defined by the quintiles. We then calculated the expected number of colonisation events in each of the five groups and compared these with the observed number of acquisitions. Within each of these groups, the posterior median of the predicted number of acquisitions was close to the observed number of events, and the observed values were always within the 80% CrI of the model estimates (*Figure 2D*). The results from fitting risk factor model A with alternative prior distributions are shown in *Figure 2—figure supplement 2*; substantially reducing the variance of the priors for the model intercept and covariates had a negligible effect on the posterior parameter estimates.

When the intercept was permitted to vary by study month in a hierarchical model (risk factor model D; *Table 2*), little variation was observed between months; the median posterior baseline probability ranged from 0.20 to 0.23 with wide credible intervals. As these models did not include a colonisation pressure term, the intercept incorporated time-varying changes in the underlying intensity of transmission. The low variance in the monthly intercepts therefore suggests a relatively constant force of infection over the 12-month study period. In models where we included a colonisation pressure term (risk factor model C; *Table 2*) this was found to have a slightly negative slope for acquisition of 3GC-R *K. pneumoniae s.l.* (OR 0.96 [95% CrI 0.86, 1.09]). This is surprising, as if patient-to-patient transmission was occurring, we would usually expect the force of infection to increase with the colonisation pressure (*Bonten, 2012*). The finding therefore suggested that one of the following three possibilities was true: i) patient-to-patient transmission was not occurring at a high frequency in this ward; ii) patient-to-patient transmission was occurring but, because of the continually high ward-level prevalence, variations in the force of infection could not be identified; iii) patient-to-patient transmission was occurring but exposure to the presence of two or more colonised patients presented a similar risk for acquisition as exposure to one. We therefore used *K. pneumoniae s.l.* whole-genome sequence data to help determine the most plausible scenario.

## *Klebsiella* whole-genome assemblies

We examined whole-genome assemblies of 317 3GC-R *K. pneumoniae s.l.* isolates cultured from rectal or environmental swabs in the neonatal unit over a four month period (see Methods). A phylogeny based on k-mer distances between assemblies is shown in *Figure 3A*. Of note is the highly diverse and structured nature of the pathogen population, in contrast to one dominated by a clonal expansion of a single lineage. Overall 62 distinct sequence types were identified in our collection of isolates. The species identified from culture as *K. pneumoniae s.l.* consists of three distinct subpopulations that meet the criteria for separate species. We isolated all three species from infants in the

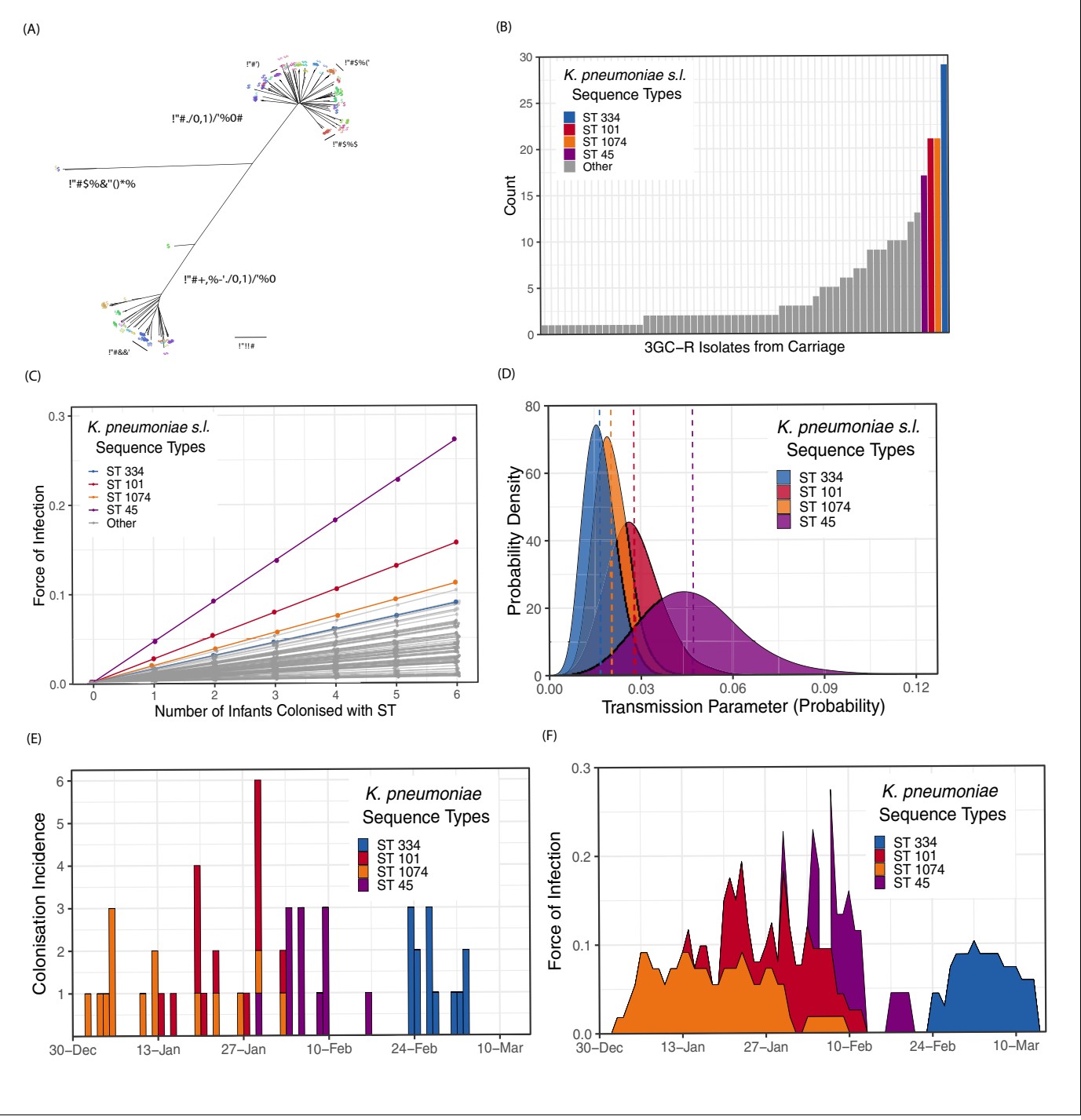

**Figure 3.** Population structure of third-generation cephalosporin-resistant (3GC-R) *Klebsiella* and force of infection by sequence type (ST). An unrooted phylogeny of 317 3GC-R *Klebsiella* isolates cultured from rectal and environmental swabs over a four month period in a neonatal unit in a children's hospital in Cambodia is shown in panel **A**, where the branch lengths correspond to the mash distance (a measure of k-mer similarity) between whole-genome assemblies. The four largest STs are labelled as well as the population subdivisions by *Klebsiella* species. The frequency distribution of STs is shown in panel **B**, with the four largest STs shown in colour. Results from a transmission model estimating the force of infection by ST are shown in panel **C**, where the force of infection scales linearly with the number of colonised infants with that ST. The largest four STs have again been highlighted. Horizontal jitter has been applied to prevent overplotting of points. The uncertainty around the transmission parameter estimates are shown in panel **D** for the four most common STs, where the posterior mean is shown with a dotted line. The daily incidence of new colonisation events with the four most frequent STs are shown between the 1st January to the 15th March 2014 in panel **E**, along with the estimated force of infection over the same period in panel **F** using parameter estimates of β from transmission model 4 (*Table 3*).

*Figure 3 continued on next page*

*Figure 3 continued*

The online version of this article includes the following figure supplement(s) for figure 3:

**Figure supplement 1.** Posterior chains from Hamiltonian Markov chain Monte Carlo fitting for *Klebsiella* transmission models (see *Table 3*).
**Figure supplement 2.** Estimation of $\lambda$ from transmission model 3 under different prior assumptions.

cohort (*K. pneumoniae* $n$ = 219, *K. quasipneumoniae* subspecies *similipneumoniae* $n$ = 95, *K. variicola* $n$ = 3), and found diversity similar to that observed in a global collection of *K. pneumoniae* isolates (*Holt et al., 2015*), suggesting that the diversity accumulated within a Cambodian neonatal unit over four months is comparable to the diversity of *K. pneumoniae* globally. Many STs were characterised by only a single carriage isolate, suggestive of importations that were not subsequently transmitted to other patients (*Figure 3B*). The STs with the largest number of carriage isolates were ST334 (*K. quasipneumoniae*; $n$ = 29), ST101 (*K. pneumoniae*; $n$ = 21), ST1074 (*K. pneumoniae*; $n$ = 21) and ST45 (*K. pneumoniae*; $n$ = 17). The most frequent $bla_{ESBL}$ genes in the whole-genome assemblies were CTX-M-15 (201/317; 63%), followed by CTX-M-14 (43/317; 13%) and CTX-M-63 (38/317; 12%).

## Transmission models for sequence types

We fitted mechanistic models representing different transmission processes to the ST swab data. Within the four month period where sequence data were available, there were 171 events for first acquisition of a 3GC-R *K. pneumoniae s.l.* ST among 150 infants. Among transmission models 1–3 that were initially tested, the model with the best fit to data by WAIC was transmission model 2 (see *Table 3* and Methods), which has an intercept ($\alpha$), representing a constant risk of acquisition, and a slope ($\beta$) which scales the risk of acquisition for each infant colonised with a given ST in the ward (i.e. it accounts for colonisation pressure). The model estimated the values for $\alpha$ as 0.0019 (95% CrI 0.0015, 0.0023) and $\beta$ as 0.0097 (95% CrI 0.0075, 0.012). We then fitted transmission model 2 with a random effect term, where $\beta$ was permitted to vary by ST and the underlying distribution of $\beta$ was assumed to follow a beta distribution, with shape hyper-parameters $\alpha$ and $\beta$ (transmission model 4).

**Table 3.** Transmission models fitted to prospectively collected, genotyped swab data on the acquisition of third-generation cephalosporin-resistant (3GC-R) *Klebsiella pneumoniae sensu lato*.

The table shows the parameters, prior and posterior distributions along with the WAIC (model comparison measure where lower values indicate a better fit to data). See methods for equations. Normal prior distributions show the mean and standard deviation respectively within brackets, beta prior distributions show the two shape parameters within brackets.

| Transmission Model (Equations in Methods) | Parameters | Priors | Posterior Median (95% CrI)[*] | WAIC [†] |
|---|---|---|---|---|
| (1) Constant risk of transmission | $\alpha$ (intercept) | beta(2, 8) | 0.0038 (0.0032, 0.0044) | 1919 |
| (2) Pseudo mass action (PMA) principal | $\alpha$ (intercept)<br>$\beta$ (transmission)[‡] | beta(2, 8)<br>beta(2, 8) | 0.0019 (0.0015, 0.0024)<br>0.0096 (0.0075, 0.012) | 1739 |
| (3) PMA plus environmental contamination, from colonised patients, which decays over time | $\alpha$ (intercept)<br>$\beta$ (transmission)[‡]<br>$\gamma$ (environment)<br>$\lambda$ (decay term) | beta(2, 8)<br>beta(2, 8)<br>beta(2, 8)<br>half-normal(1, 2) | 0.0019 (0.0014, 0.0023)<br>0.0088 (0.0063, 0.011)<br>0.097 (0.0086, 0.37)<br>4.7 (2.2, 7.1) | 1740 |
| (4) Hierarchical PMA varying transmission coefficient by ST[4] | $\alpha$ (intercept)<br>$\beta$ (transmission)[‡]<br>$\alpha_s$ (beta shape 1)<br>$\beta_s$ (beta shape 2) | beta(2, 8)<br>beta($\alpha_s$, $\beta_s$)<br>half-normal(2, 5)<br>half-normal(8, 5) | 0.0021 (0.0016, 0.0026)<br>Varies by ST[§]<br>0.23 (0.14, 0.38)<br>17 (9.7, 25) | 1733 |
| (5) Hierarchical PMA varying intercept by ST[§] | $\alpha$ (intercept)<br>$\beta$ (transmission)[‡]<br>$\alpha_s$ (beta shape 1)<br>$\beta_s$ (beta shape 2) | beta($\alpha_s$, $\beta_s$)<br>beta(2, 8)<br>half-normal(2, 5)<br>half-normal(8, 5) | Varies by ST[§]<br>0.0094 (0.0073, 0.012)<br>0.23 (0.16, 0.31)<br>21 (13, 29) | 1793 |

[*] 95% Credible interval. [†] Widely applicable information criterion (WAIC). [‡] Transmission parameter that is multiplied by the number of infants colonised with the sequence type on the same day to give the force of infection (colonisation pressure). [§] 3GC-R *K. pneumoniae s.l.* sequence type (ST).

We also fitted transmission model 2 with a random effect term where α was permitted to vary by ST (transmission model 5).

Transmission model 1 has a constant colonisation pressure by ST which was not linked to the daily number of colonised individuals with that ST; this model showed a substantially worse fit to the data by WAIC (*Table 3*). Transmission model 3, which included terms for colonisation pressure and contamination in the hospital environment also failed to improve the model fit, and the high estimate for λ (4.7 95% CrI [2.0, 7.0]) suggests that contamination left by previously colonised infants decays rapidly to background levels, with an estimated environmental half life of 3.6 hours (95% CrI 2.4, 7.6 hours; *Table 4*). Varying the α parameter by ST (transmission model 5) resulted in a substantially worse fit to data, suggesting there was not enough information in the model to differentiate ST-specific background rates of colonisation.

The central estimates of the force of infection by ST from transmission model 4 are shown in *Figure 3C*, with estimates from the four most frequent STs highlighted in colour. The uncertainty around these parameter estimates for the four STs is shown in *Figure 3D*. The daily interval-censored colonisation incidence for the most frequent STs are clustered in time, generally emerging and reaching extinction in the ward within a matter of weeks, suggestive of importation and subsequent patient-to-patient transmission. The incidence and estimated force of infection for the four most frequent STs over a period in the study where all 3GC-R *K. pneumoniae s.l.* isolates were sequenced are shown in *Figure 3E and F*. Parameter estimates for all transmission models are shown in *Table 3*. Model fitting diagnostics showed that Markov chains had converged satisfactorily (see Methods and *Figure 3—figure supplement 1*).

## Sequence Type SNP diversity

We mapped reads from isolates in two STs with the highest estimated force of infection (ST45 and ST101; see *Figure 3*) to ST consensus reference genomes. The 21 ST101 carriage isolates had a mean read depth ranging from 25x to 125x (median 57x), and the 17 ST45 carriage isolates had a mean read depth ranging from 20x to 67x (median 56x). All the carriage isolates in both STs had >90% of the genome covered by >5x coverage. We then called and filtered SNPs (see Methods) to determine if the relatedness of carriage isolates within STs was consistent with recent person-to-person transmission. The pairwise number of variants in ST101 isolates between infants ranged from 15 to 68 SNPs (median 30), which was comparable to the variation seen within infants in ST101 (from 18 to 38 SNPs; median 28). Similarly in ST45 the pairwise SNP differences between infants varied from

**Table 4.** Key epidemiological parameters estimated in this study from longitudinal swab data on third generation cephalosporin-resistant *Klebsiella pneumoniae sensu lato* from a neotatal intensive care unit from a Children's Hospital in Cambodia.

| Parameter | Method | Estimate | Uncertainty interval | Key Assumptions |
|---|---|---|---|---|
| Daily risk of acquisition for neonates | Bayesian regression model | 0.15 | 0.091, 0.19 (IQR[*]) | Culture diagnostic 100% sensitive |
| Force of infection from one colonised infant | Bayesian transmission model | 0.016 | 0.0093, 0.027 (95% CrI[†]) | Expected values from transmission model 4 |
| Swab sensitivity (1) | Negatives following a positive swab Beta conjugate prior | 0.90 | 0.88, 0.92 (95% CrI[†]) | All positives are false negatives Beta(1,7) prior |
| Swab sensitivity (2) | Negatives following a positive swab Beta conjugate prior | 0.93 | 0.91, 0.94 (95% CrI[†]) | Three consecutive negatives are a true decolonisation Beta(1,7) prior |
| Environmental half life[‡] | Bayesian transmission model | 3.6 hours | 2.4, 7.6 hours (95% CrI[*]) | Exponential decay Normal(1, 2) prior |
| Ward reproduction number $R_A$) | Agent-based simulation | 0.65 | 0.36, 1.1 (95% interval[§]) | Ward size of 8 susceptible neonates |

[*] Interquartile range (IQR) taken from distribution of daily risk of acquisition (**Figure 2B**). [†] Credible interval (CrI). [‡] Inverse rate of decay of environmental contamination, as estimated in transmission model 3, multiplied by ln(2). [§] 95% of simulated values fell within this interval.

13 to 223 (median 125), which was comparable to the within-host ST diversity (from 55 to 212 SNPs; median 124). Therefore the SNP diversity observed within and between-hosts was very similar for both ST45 and ST101.

## Simulations with agent-based models

We used the posterior parameter estimates obtained from model fitting for forward simulations using a dynamic agent-based model in order to evaluate the potential impact of interventions (see Methods). We first estimated the ward-level reproduction number ($R_A$) for 3GC-R *K. pneumoniae s.l.* by simulating the introduction of a single colonised patient into a ward of eight susceptible patients. For all patients, length of stay was sampled from the empirical length of stay distribution (*Figure 1C*) and colonised patients had a transmission potential sampled from the posterior hyper-parameter distribution from transmission model 4 (*Table 3*). The median of the $R_A$ distribution was 0.65% and 95% of values fell between 0.36 and 1.09. The distribution of $R_A$ values is shown in *Figure 4A*.

We then simulated the impact of interventions to reduce the rate of 3GC-R *K. pneumoniae s.l.* acquisition by combining parameter estimates for colonisation pressure from transmission model 4 with the marginal effect of modifiable covariates from risk factor model A (*Table 2*). In the first intervention scenario, we varied the proportion of infants given an oral probiotic, in addition to varying the proportion of infants that were colonised on entry (imported cases; 5% or 40%). We used as an outcome the proportion of infants susceptible to 3GC-R *K. pneumoniae s.l.* on admission that remained uncolonised on discharge. When the importation rate was high (40% colonised on entry; similar to our study population, see *Table 1*), setting the proportion of infants taking the probiotic to be 0%, 50% or 100% resulted in the median proportion remaining uncolonised as 0.54 (95% interval 0.34, 0.72), 0.56 (95% interval 0.38, 0.72) and 0.59 (95% interval 0.39, 0.75) respectively. In the lower importation setting (5% of infants colonised on entry), setting the proportion of infants taking the probiotic to be 0%, 50% or 100% resulted in the median proportion remaining uncolonised as 0.80 (95% interval 0.54, 0.93), 0.82 (95% interval 0.60, 0.93) and 0.84 (95% 0.61, 0.94), respectively.

In the second intervention scenario, we varied the proportion of breast milk fed infants between 25%, 50% and 90%, in addition to varying the proportion of infants that were categorised as imported cases. When the importation rate was high (40%), altering the proportion of infants breast fed between 25%, 50% and 90% resulted in the median proportion remaining uncolonised as 0.48 (95% interval 0.25, 0.69), 0.51, (95% interval 0.31, 0.70), and 0.56 (95% interval 0.38, 0.72), respectively. When the importation rate was low (5%) altering the proportion of infants breast fed between 25%, 50% and 90% resulted in the median proportion remaining uncolonised as 0.73 (95% interval 0.40, 0.92), 0.77 (95% interval 0.48, 0.92) and 0.84 (95% 0.61, 0.94) respectively.

In the third scenario, we simulated either three, four or eight nurses in the ward each day, corresponding to infant:nurse ratios of roughly 3:1, 2:1 and 1:1 respectively. Again, we examined this effect in settings with different importation rates. Varying the number of nurses in the ward between three, four and eight in the high importation setting resulted in median proportions of initially uncolonised infants who remained uncolonised throughout their neonatal unit stay of 0.54 (95% interval 0.36, 0.71), 0.61 (95% interval 0.39, 0.78) and 0.81 (95% interval 0.21, 0.97) respectively. In the lower importation setting, varying the infant:nurse ratio resulted in the median proportions remaining uncolonised of 0.81 (95% interval 0.56, 0.93), 0.86 (95% interval 0.62, 0.95) and 0.96 (95% interval 0.32, 0.99), respectively. Of all the simulated interventions therefore, increasing the number of nurses on the ward had the largest impact on reducing colonisation rates. The distributions of the outcome variables from all simulations are shown in *Figure 4B, C and D*.

## Discussion

In a hyperendemic, developing country hospital setting, we analysed the transmission dynamics of 3GC-R *K. pneumoniae s.l.* prospectively over one year. We compared the support for competing hypotheses about modes of spread, quantified effects of antibiotic exposures as drivers of the epidemic, evaluated risk factors for transmission, and forward simulated to evaluate the potential benefits of interventions.

We found that carriage strains of 3GC-R *K. pneumoniae s.l.* among neonates constituted a highly diverse population, with considerable intra-host variation within STs. There were frequent

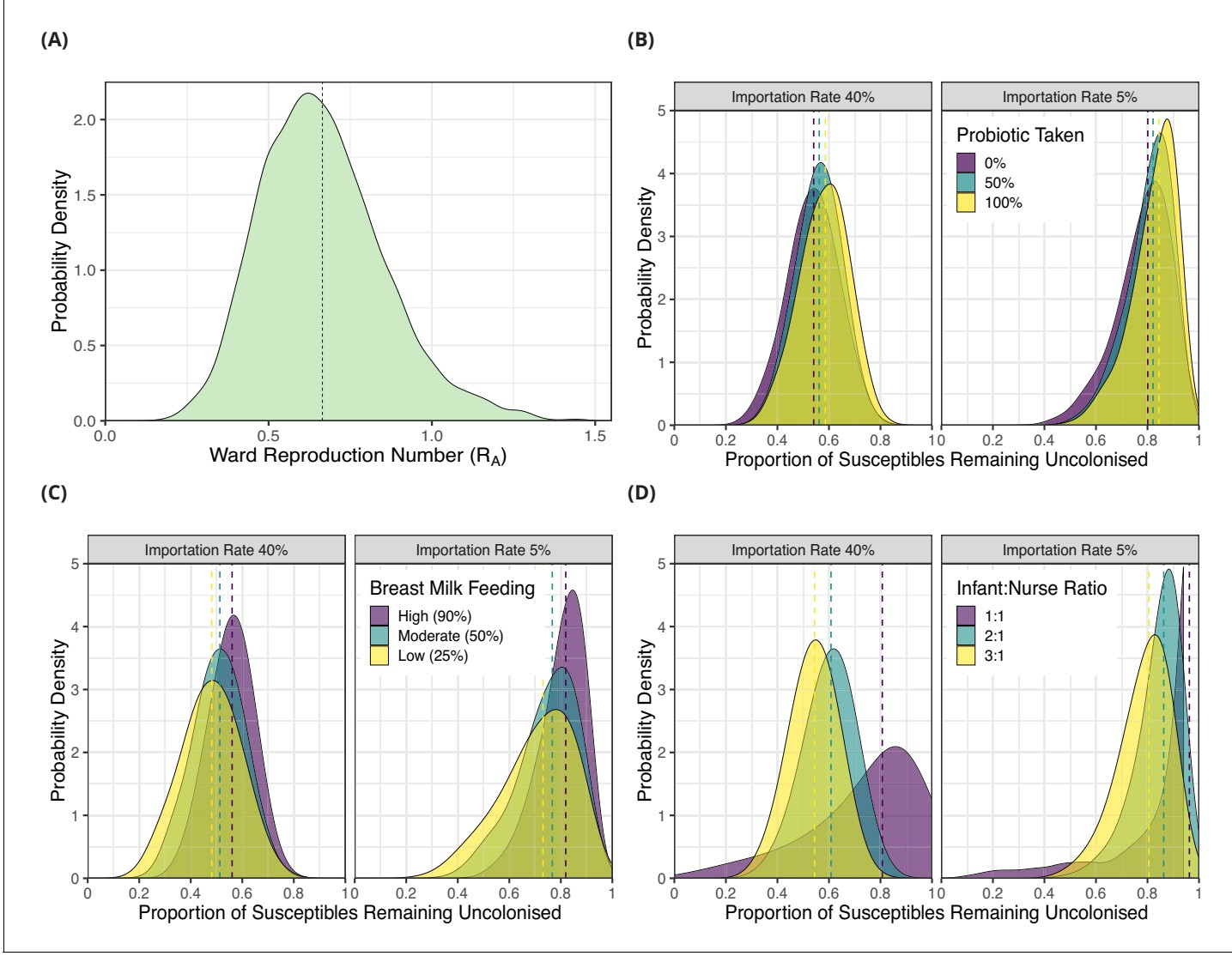

**Figure 4.** Simulation results from dynamic agent-based models using parameter estimates on acquisition of third generation cephalosporin-resistant (3GC-R) *Klebsiella pneumoniae sensu lato* among neonates in a Children's Hospital in Cambodia. The distribution of ward reproduction number (R_A) values shown in panel **A** was obtained by taking 2000 samples from the force of infection posterior distribution, and for each sample running the agent-based simulation 100 times and taking the mean value. The results from simulating counterfactual scenarios with a dynamic agent-based model are shown in panels **B**, **C** and **D**. In B, the proportion of infants taking a probiotic (Lactobacillus acidophilus) on entry to the ward was varied between 0, - .5 and 1 in setting with a high proportion of imported cases (0.4) and a lower proportion of imported cases (0.05). In panel C, the proportion of infants that were breast milk fed was varied was varied between 0.25, 0.5 and 0.9in settings with a high proportion of imported cases (0.4) and a lower proportion of imported cases (0.05). In panel D, the infant nurse ratio was varied between 3:1, 2:1 and 1:1 in settings with a high proportion of imported cases (0.4) and a lower proportion of imported cases (0.05). The outcome measure in all simulations is the proportion of infants susceptible on entry that remained uncolonised with 3GC-R K. pneumoniae s.l. on discharge. The simulated outcomes are displayed as density plots, with dashed lines showing the median value.

acquisitions or detection of resistant *K. pneumoniae s.l.* that were closely related to strains carried by other infants on the ward at the time of acquisition (*Figure 3E*). Within-host diversity of these lineages was similar to between-host diversity in potentially linked cases. Moreover, once genomic information was considered, models incorporating colonisation pressure as a risk factor for acquisition showed substantially better fits to data than models without colonisation pressure (*Table 3*). Taken together with the lack of persistent environmental contamination (*Smit et al., 2018*) and the lack of improvement in model fit when long-term environmental contamination was considered, these findings add support to the view that patient-to-patient transmission (much of which is likely

to be mediated by contacts with healthcare workers) is the main driver of resistant *K. pneumoniae* acquisition, and that colonised patients represent the primary reservoir. Notably, this result was only apparent when we incorporated genomic data into our models to estimate transmission parameters by ST.

We are aware of one other prospective study of ESBL-producing Enterobacteriaceae colonisation dynamics from a developing country (*Bonneault et al., 2019*). This study considered ESBL-producing Enterobacteriaceae colonisation in a neonatal unit in Madagascar over a period of six months, and found similar rates of acquisition with ESBL-producing Enterobacteriaceae to those seen here. Though sequencing data were not used, fitting transmission models to data provided evidence of patient-to-patient and healthcare worker-patient transmission, particularly for ESBL-producing *K. pneumoniae* and the estimated daily transmission parameter (0.05; 0.008, 0.14) is similar to our estimates for the most common STs (*Figure 3D*).

Two other prospective carriage studies of *K. pneumoniae* have used whole genome sequencing to identify possible transmission events: while a one year study in an adult intensive care unit in Australia found five epidemiologically plausible intra-hospital transmission chains (*Gorrie et al., 2017*), a one year carriage study in two geriatric wards (also in Australia) found no evidence of patient-to-patient transmission chains (*Gorrie et al., 2018*). Other studies using genomic epidemiology to study resistant *K. pneumoniae* transmission have been performed in adult wards in high-income settings (*Snitkin et al., 2012*; *Haller et al., 2015*; *Snitkin et al., 2017*). While such studies have also provided support for patient-to-patient transmission playing an important role, such retrospective investigations made in response to reported outbreaks of multidrug-resistant *K. pneumoniae* might not reflect typical patterns of transmission. A study using proximity sensors to investigate transmission of ESBL-producing Enterobacteriaceae over four months in a long term care facility in France found stronger support for person-to-person transmission as the main route of acquisition for *K. pneumoniae*, though the evidence for person-to-person transmission of ESBL-producing *E. coli* was weaker (*Duval et al., 2019*). This suggests that separate mechanisms may drive the transmission of different ESBL-producing organisms and that this should be considered when analysing carriage data from multiple species of Enterobacteriaceae.

Under the assumption that patient-to-patient transmission was driving the epidemic, we calculated that the rates of transmission within the ward were insufficient to maintain endemic transmission (i.e. the ward-level reproduction number, $R_A < 1$) (*Cooper et al., 2004*). These findings are comparable to a study which determined the relative force of infection between ESBL-producing *E. coli* and other ESBL-producing Enterobacteriaceae in 13 European intensive care units, finding that the latter (mainly *K. pneumoniae*) had a transmission rate almost three times greater than the former, and that the single-admission reproduction number for both classes of organisms were well below one (*Gurieva et al., 2018*). Our central estimate of $R_A$ is, however, substantially higher (0.65 compared with 0.17); this difference in estimated transmission potential may reflect differences in staff-to-patient ratios, different standards of hygiene and infection control, different patterns of antibiotic use, and differences in the patient population (*Dondorp et al., 2018*). The findings from both studies indicate that repeated importation into the unit is needed to maintain endemicity of resistant *Klebsiella*. Imported cases may be acquired from other wards within the same hospital, other hospitals within the referral network, or community transmission. When rates of importation are high, as observed in this setting where up to 43% of infants may have been colonised on initial ward admission (*Table 1*), even effective interventions are limited in how many acquisition events they can prevent due to a high underlying colonisation pressure (*Figure 4*). The public health implications are therefore that control measures should be coordinated regionally and targeted to the wider hospital patient referral network (*Donker et al., 2012*; *Ciccolini et al., 2013*).

Amongst the most important findings was the consistent association between a patient's antibiotic exposure and an increased risk of acquiring 3GC-R *K. pneumoniae s.l.* or detection of the organism due to within-host selection. To our knowledge, the role of antibiotics as drivers of carriage dynamics has not previously been explored with appropriate methods to account for time-dependent antibiotic exposures in Gram-negative bacteria, but there are reasons for believing it is likely to be a key mechanism through which antibiotics select for ESBL *K. pneumoniae* (*Tedijanto et al., 2018*). With the exception of imipenem, for which there was no association with the risk of acquisition for these predominantly carbapenem-susceptible bacteria (1418/1423 *K. pneumoniae s.l.* isolates sensitive to imipemem; 99.6%), effect sizes were similar for different antibiotic combinations

(median posterior OR ~2; *Figure 2A*). The narrower credible interval for the ampicillin + gentamicin combination (OR 1.96, 95% CrI 1.18, 3.36) reflects the much higher usage of this antibiotic combination compared to others (*Figure 1D*). These effects are consistent with hypotheses about antibiotic therapy leading to reduced microbiome diversity and subsequently increasing the risk of colonisation with drug-resistant bacteria, which face less competition from fitter, sensitive strains (*Lipsitch et al., 2000*). Microbiome analysis has shown that the greatest dysbiosis following antibiotic therapy in healthy adults is four days after treatment starts (*Palleja et al., 2018*), lending confidence to our model comparison selecting a 96 hour exposure period over a 48 hour period (*Table 2*). While we cannot differentiate the effects of antibiotics on i) increasing the susceptibility of an infant to acquisition from another infant, versus ii) endogenous selection for resistant bacteria within that infant (*Lipsitch and Samore, 2002*) in the risk factor models, the small estimate for α relative to β in transmission model two suggests that background selection plays a relatively small role in acqustion compared with person-to-person transmission (*Table 3*).

Breast milk feeding was associated with a reduced risk of colonisation with 3GC-R *K. pneumoniae s.l.* though uncertainty was large (*Figure 2A*). This accords with our understanding of the development of a healthy gut microbiome in the early stages of life, which can be adversely affected by the replacement of breast milk by formula (*Bäckhed et al., 2015*), and the protective effect of a diverse microbiome that competes against potentially pathogenic bacteria (*Langdon et al., 2016*). In this population, breastfeeding rates were high (90%) though our simulations showed that dropping the proportion to 50% or 25%, as seen in other developing world populations (*Lauer et al., 2004*), could increase the proportion becoming colonised during admission by around 5% and 8% respectively.

The finding that the oral probiotic *Lactobacillus acidophilus* was not strongly protective against acquisition was disappointing in light of earlier results that suggested a possible effect in slowing rates of acquisition of ESBL-producing Enterobacteriaceae (*Turner et al., 2016*). This negative result was shown by the odds ratio, which was located close to zero in the risk factor models for acquisition (*Figure 2A*) and supported by the forward simulations which showed only a 4% median decrease in infant colonisation rates when 100% of infants were prescribed probiotics (*Figure 4B*). Evidence is still limited about the value of probiotics for neonates. One large randomised trial in rural India reported a beneficial effect from a symbiotic preparation (combining a probiotic, *Lactobacillus plantarum*, with fructooligosaccharide) in preventing sepsis in infants (*Panigrahi et al., 2017*). Another randomised trial in pre-term infants in England found no benefit from the probiotic *Bifidobacterium breve* BBG-001 in preventing necrotising enterocolitis, blood culture positive sepsis or death before hospital discharge (*Costeloe et al., 2016*). There is also evidence that some probiotics given after antibiotic consumption can impair and delay the recovery of normal gut flora in humans (*Suez et al., 2018*).

An increased number of nurses on the ward was negatively associated with the risk of acquiring 3GC-R *K. pneumoniae s.l.*. Our simulation studies using a dynamic agent-based model showed that increasing the nurse:infant ratio from 1:3 to 1:1 could reduce the number of infants becoming colonised in both high and low importation settings by about a quarter. These findings are consistent with results from large observational studies. For example, (*Rogowski et al., 2013*), in a retrospective cohort study in 67 neonatal units in the USA found a strong association between neonatal unit understaffing and an increased nosocomial infection rate (where understaffing was defined as a nurse-patient ratio below US guidelines for the patient acuity level). Two distinct mechanisms might account for such an association. First, an imbalance between workload and staffing levels may lead to reduced attention to basic infection control measures such as hand hygiene, as has been reported in several studies (*Pittet et al., 2006*). Second, as nurse:patient ratios decrease a lower proportion of patient contacts will be cohorted as each nurse will need to contact more patients in a shift, substantially increasing the potential for cross-transmission (*Archibald et al., 1997*; *Austin et al., 1999*; *Hugonnet et al., 2004*).

From our model estimates, we observed considerable variation in the risk of acquisition per patient day, with the median posterior probability varying nearly eight-fold from 0.047 to 0.35, showing that even neonates within a single hospital ward constitute a highly heterogeneous population. This challenges the assumptions of compartmental models for nosocomial transmission that treat patients as broadly homogeneous in their risk of acquiring drug-resistant bacteria (*Grundmann and Hellriegel, 2006*; *van Kleef et al., 2013*; *Domenech de Cellès et al., 2013*).

The effect of colonisation pressure was not identifiable when we considered the total number of 3GC-R *K. pneumoniae s.l.* isolates on a given day, but became identifiable when we considered transmission by ST. From our analysis of the *Klebsiella* carriage population, we observed three species (*K. pneumoniae*, *K. quasipneumoniae* and *K. variicola*) along with a considerable number (62) of STs. This indicates that *Klebsiella* in carriage did not reflect a single population, but rather repeated introductions of diverse isolates which were then either spread around the ward over a number of weeks or were not transmitted and became locally extinct.

Comparison of different transmission models strongly supported the inclusion of a colonisation pressure term to account for patient-to-patient transmission. There was also support for a hierarchical model where transmissibility varied by ST, though the STs we estimated to have the highest transmissibility have not been highlighted as dominant in other settings (*Wyres and Holt, 2016*) suggesting that the ST composition within a region may reflect adaptation to local pressures (*Stoesser et al., 2015b*). The pairwise SNP diversity in two of the major STs (ST45 and ST101) was greater than expected, but the within-host and between-host diversity was similar in both cases, suggesting that transmission within these STs is biologically plausible. Within-host diversity has the potential to hinder the reconstruction of transmission networks (*Worby et al., 2014*; *Didelot et al., 2016*), and our results here highlight the importance of capturing within-host diversity in sequencing studies. While within-host diversity was particularly high in ST45, all acquisitions of this ST occurred within a 19 day window (over a possible four month period when carriage isolates were sequenced) with overlapping colonised patient stays, strongly suggesting that cases were epidemiologically related (*Figure 3E*). A previous analysis of a subset of the genomic data identified closely related clusters suggestive of transmission, though with a smaller number of pairwise SNPs than we observed (*Smit et al., 2018*). This discrepancy is likely due to differences in methodology as we mapped isolates to ST-specific reference genomes, which results in a greater proportion of the genome being callable compared with mapping very diverse isolates to a single reference. We cannot entirely rule out less parsimonious explanations for the temporal clustering of STs, such as a transient increase of certain STs in the water supply, though the person-to-person transmission route, mediated by healthcare workers, is most strongly supported by our models.

An important strength of the study is that we considered asymptomatic carriage, rather than clinical isolates, and collected detailed patient-level data from infants who became colonised as well as from infants who remained uncolonised. This allowed us to quantify the factors driving the epidemic, which would not have been possible if we had considered only clinical isolates. Also, by using a prospective design rather than a reactive exploration of an outbreak or unusual cluster of cases our findings should be more representative of typical patterns of transmission. The use of an inferrential approach that accounted for the interval-censored nature of the data and the use of whole genome sequencing were also key factors in developing a quantitative understanding of the transmission dynamics.

Our study has limitations. We sequenced isolates of 3GC-R *K. pneumoniae s.l.* from a four month period (rather than the full 12 months of the carriage study) due to resource constraints. Such constraints also prevented us from sampling mothers and other family members of infants, and healthcare workers in close contact with infants. Greater density of within-host sampling (*Stoesser et al., 2015a*; *Wymant et al., 2018*; *Lees et al., 2019*), and inclusion of long-read sequencing to investigate plasmid transmission (*Conlan et al., 2014*) would have also provided a more complete epidemiological picture. Although our analysis accounts for patient heterogeneity and interval censoring, it assumes that a positive culture taken within 48 hours of admission indicates that the patient was colonised on admission, and that cultures had 100% sensitivity for detecting carriage. Such restrictions could all be relaxed using a data augmentation framework (*Cooper et al., 2008*), but validated implementations of such an approach allowing inclusion of an arbitrary number of covariates are not currently available. Antibiotics were not prescribed randomly, but according to the clinical judgement of the clinicians and in response to the pathology of the patient. It is therefore possible that the effects of antibiotics in the model estimates may be confounded by the clinical severity of the infant. We did attempt to mitigate this by including a covariate for clinical severity, which links to the use of invasive devices and we note that, with the exception of imipenem, all antibiotics show similar effects to each other on the risk of first acquisition.

In summary, this study provides strong evidence for person-to-person within ward transmission of 3GC-R *K. pneumoniae s.l.* mediated by healthcare workers, estimates key epidemiological

parameters, quantifies the role of different antibiotics in driving the epidemic, and highlights interventions with the potential to contribute to control efforts.

## Materials and methods

### Epidemiological and microbiological data

We used data collected prospectively from a neonatal intensive care unit in a children's hospital in Siem Reap, Cambodia between September 2013 (when the neonatal unit newly opened) and September 2014. The study protocol required rectal swabs to be taken within 48 hours of admission and subsequently every two to three days. We assumed that any patient testing positive on their first swab taken within 48 hours of admission was positive on arrival. Patients who had their first swab taken >48 hours from admission but tested negative were included in the analysis, while those that tested positive were omitted. Infants that were re-admitted to the ward and had been colonised prior to first discharge were assumed to still be colonised and were thus excluded. Details of the microbiological treatment of rectal swabs, including resistance assays, have been published previously (*Turner et al., 2016*).

### Probability of acquisition

We estimated the daily probability of colonisation with 3GC-R *K. pneumoniae s.l.* using a discrete time model with time steps of one day. Each day in the ward a previously uncolonised patient can become colonised. As rectal swabs are not taken on every day of a patient's stay the outcome is interval censored: we know that a negative swab followed by a positive swab indicates that a patient became colonised on some day between the two swabs, but not on which day. If the probability of becoming colonised on day $i$ for patient $j$ is $p_{ij}$, given the patient is uncolonised at the start of the day, then the probability of remaining uncolonised is $(1-p_{ij})$. In interval $k$ for patient $j$ consisting of $N_{kj}$ days, then the probability of remaining uncolonised is:

$$\prod_{i=1}^{N_{kj}}(1-p_{ij})$$

Therefore the probability of becoming colonised ($v_{kj}$) is the complement:

$$v_{kj} = 1 - \prod_{i=1}^{N_{kj}}(1-p_{ij})$$

The outcome for patient $j$ in interval $k$, $Y_{kj} \in \{0,1\}$, as the patient either becomes colonised (1) or remains uncolonised (0) with 3GC-R *Klebsiella*. Therefore the likelihood is given by:

$$Y_{kj} \sim Bernoulli(v_{kj})$$

### Risk factor models for carriage acquisition

The daily probability of becoming colonised ($p_{ij}$) is related by the logit link function to a linear function of covariates:

$$\pi_{ij} = \alpha + \beta_1 x_1 + \beta_2 x_2 + \beta_3 x_3 ...$$

$$p_{ij} = \frac{exp(\pi_{ij})}{exp(\pi_{ij}) + 1}$$

Where $x_1$, $x_2$, $x_3$ ... is a vector of predictors (data) and $\beta_1$, $\beta_2$, $\beta_3$ ... is a vector of slopes (parameters) that are to be estimated. The intercept $\alpha$ can be a single parameter, or permitted to vary over $m$ periods of time in a random effects model. When using such a random effects model $\alpha$ was assumed to be normally distributed, with a mean $\mu$ and standard deviation $\sigma$, which are themselves parameters with their own prior distributions. The prior distributions used in the analysis are shown in *Table 2*. Results from models with alternative prior distributions are shown in *Figure 2—figure supplement 2*.

$$\alpha_m \sim normal(\mu, \sigma)$$

Fourteen standard covariates included in all risk factor models were: exposure to the six most common combinations of antibiotics, taken intravenously unless otherwise stated (ampicillin, ampicillin + gentamicin, cloxacillin (oral), ceftriaxone, cloxacillin + gentamicin, and imipenem) within the past 48 or 96 hours; whether the infant was breast milk fed; if the infant recieved an oral probiotic on entry (*Lactobacillus acidophilus*) for prevention of necrotising enterocolitis; sex; born prematurely (before the 37th week of pregnancy); severity (defined as either i. requiring ventilation, ii. requiring continuous positive airway pressure or iii. requiring inotopes); and if already colonised with 3GC-R *E. coli*. These explanatory variables were treated as binary (0/1). We also included the age in days on first admission to the NU, and the daily number of nurses on the ward. An additional covariate included in one of the risk factor models (see below) was a term for colonisation pressure, which is an integer value representing the number of individuals known to be colonised with 3GC-R *K. pneumoniae s.l.* on that day. Covariates were recorded for every day the infant was present in the neonatal unit and data were treated as complete. We considered the following models:

A. Single intercept with standard covariates (14). Exposure to antibiotics considered if taken within the past 96 hours.
B. Single intercept with standard covariates (14). Exposure to antibiotics considered if taken within the past 48 hours.
C. Single intercept with standard covariates (14) plus an additional covariate for colonisation pressure. Exposure to antibiotics considered if taken within the past 96 hours.
D. Variable intercept by study month that uses partial pooling (hierarchical model). Standard covariates (14) and exposure to antibiotics considered if taken within the past 96 hours.

## Rectal swab/culture sensitivity

Swab sensitivity was estimated from the number of negative swabs following a positive swab, i) under the assumption that all negative results following a positive swab were false negatives and ii) under the assumption that three or more consecutive negative swabs following a positive represented a true decolonisation event. Posterior distributions of the false negative rate and swab sensitivity were estimated using a conjugate beta prior; beta($\alpha$=1, $\beta$=7) (*Bolker, 2008*).

## Pathogen sequencing and bioinformatics

We whole-genome sequenced 317 cultured isolates identified morphologically as 3GC-R *K. pneumoniae s.l.* from i) rectal swabs from all colonised patients within a four month period of the study and ii) twice weekly swabs from seven environmental surfaces around the ward (6 sinks and one computer keyboard) within the same time frame. Sequencing was performed with the Illumina HiSeq 2500 platform, producing 150 base-pair paired-end reads. The reads were trimmed for adapter sequence using TrimGalore (v0.4.4) before assembly with Unicycler (v0.4.5) (*Wick et al., 2017*), contigs <1 kilobase were discarded. Distances between assemblies were calculated using mash (v1.1) (*Ondov et al., 2016*) and a phylogeny constructed with mashtree (v0.33). Sequence types (STs) were identified using Kleborate 0.2.0 (*Wyres et al., 2016*). Variant calling was performed within STs by mapping reads to ST consensus reference genomes (5.32 Mbp) with SMALT (v.0.7.6) https://www.sanger.ac.uk/science/tools/smalt-0 (parameters -x -y 0.85 r 1 j 100 -i 800), before sorting and removing unpaired mate reads and technical duplicates from binary alignment files with samtools (v.1.8). Single nucleotide variants (SNPs) were called by piping output from samtools mpileup into bcftools (v.1.8) and were conservatively filtered to remove SNPs within 50 bp of indels, with a read depth <10x and >200x, a mapping quality score <30 or read quality score <100. Repetitive regions were identified with nucmer (v.3.1), phage regions with PHASTER (*Arndt et al., 2016*), and recombination with ClonalFrameML (v.1.11) (*Didelot and Wilson, 2015*), and these regions subsequently masked from SNP calling.

## Transmission models incorporating sequence type data

We assessed within-ward transmission of 3GC-R *K. pneumoniae s.l.* STs under the assumption that individuals remain colonised with a given ST for the duration of their stay until discharge (*Birgand et al., 2013*). We fitted data to five linear transmission models.

1. Where the daily risk of acquiring any ST is constant (intercept only).
2. A constant term plus a covariate for colonisation pressure ($\beta$), where the explanatory variable ($n_{ic}$) is the number of other patients colonised with ST $c$ in the neonatal unit on day $i$.
3. As (2) with an additional term for the contribution of environmental contamination to transmission ($\gamma$), whereby the number of cases with ST $c$ on an earlier day $i'$ ($n_{i'c}$) are assumed to leave some trace in the environment that decays exponentially over time at a rate given by $\lambda$.
4. A hierarchical version of (2) which permits the transmission parameter $\beta$ to vary by ST.
5. A hierarchical version of (2) which permits the intercept $\alpha$ to vary by ST.

The probability $p$ of colonisation for individual $j$ on day $i$ with ST $c$ for the respective models are:

$$p_{ijc} = \alpha \qquad \text{(Transmission Model 1)}$$

$$p_{ijc} = \alpha + \beta n_{ic} \qquad \text{(Transmission Model 2)}$$

$$p_{ijc} = \alpha + \beta n_{ic} + \gamma \sum_{i'=0}^{i-1} n_{i'c} e^{-\lambda(i-i')} \qquad \text{(Transmission Model 3)}$$

$$p_{ijc} = \alpha + \beta_c n_{ic} \qquad \text{(Transmission Model 4)}$$

$$p_{ijc} = \alpha_c + \beta n_{ic} \qquad \text{(Transmission Model 5)}$$

We opted to fit transmission models on the linear, rather than logistic, scale as we consider a linear increase in the force of infection with the number of colonised infants to be a more realistic assumption than the, initially, exponential increase that results from a logistic transformation.

## Statistical model fitting

We fitted the statistical models using Hamiltonian Markov chain Monte Carlo in Stan (version 2.17.3) within the R environment (v. 3.4.3). Prior distributions were selected to be weakly informative normal distributions for the risk-factor models (*McElreath, 2018*), see *Table 2*. For the transmission models, beta distributions were used as priors. in the case of the hierarchical transmission models (4 and 5) the scale and location parameters are themselves hyper-parameters with their own priors. Studies have shown the half life of carbapenem-resistant *K. pneumoniae* inoculated onto hospital surfaces to be around 12 hours (*Weber et al., 2015*); this corresponds to $\lambda$=1.39 in our model. We therefore gave $\lambda$ in transmission model 3 a more informative prior, normal($\mu$=1, $\sigma$=2), to give weight to biologically plausible estimates. Prior distributions for all transmission models are shown in *Table 3*. Results from a less informative prior distribution for $\lambda$ are shown in *Figure 3—figure supplement 2*. Chains were run for a varying number of iterations depending on the number of parameters to estimate, though with a minimum of 12,000 iterations over four chains, including burn-in. The Gelman-Rubin statistic ($\hat{R}$) was used as a diagnostic, where values <1.01 indicate chains have converged and additionally posterior chains were visually inspected for convergence. Effective sample sizes (ESS) for both the centre of the posterior distribution (bulk) and the ends of the distribution (tail) was ensured to be >400 (*Vehtari et al., 2019*). Model comparison was performed with widely applicable information criterion (WAIC) (*Vehtari et al., 2017*). We used 95% credible intervals (CrIs) as a measure of uncertainty around posterior parameter distributions and posterior medians as the central estimate. Uncertainty in parameter estimates is represented by the posterior distributions, and the estimated probability that a particular parameter is within a certain range is given by the area under the curve of that parameter's marginal posterior distribution within that range.

## Agent-based forward simulations

We forward simulated the impact of interventions using an agent-based model implemented in Python (v. 2.7.15) and hosted at https://github.com/tc13/ward-infection-ABM (copy archived at https://github.com/elifesciences-publications/ward-infection-ABM). In brief, we model in discrete time steps a ward containing a fixed number of beds and where patients sample a length of stay

and colonisation status on entry. Simulations to estimate the ward reproduction number $R_A$ introduced a single colonised individual into a full ward of susceptible patients. The probability of any uncolonised patient acquiring resistant *Klebsiella* from the index patient on day one is $p_{ijc}$, where $p_{ijc}$ is sampled from a beta distribution with shape hyper-parameters $\alpha$ and $\beta$ from transmission model 4 (*Table 3*). This probability changes on subsequent days based on the number of other infants that become colonised and start transmission, and the lengths of stay. The simulation ends when the index patient is discharged.

For simulations that explored the impact of interventions, the effect of each of the covariates of interest (probiotic consumption, breast milk feeding and number of nurses on the ward) were obtained from risk factor model A (*Table 2* and *Figure 2A*). The marginal effects of each risk factor were transformed into a log-odds ratio and used to modify the daily risk of acquisition sampled from a beta distribution with shape hyper-parameters $\alpha_s$ and $\beta_s$. For $n_c$ colonised patients on the ward on day $j$, the probability of an uncolonised patient with covariate $k$ acquiring 3GC-R *K. pneumoniae s.l.* is $1-(1- p_k)^{n_{cj}}$. The simulation ends after 365 days. To reduce variability between model runs for estimation of $R_A$ and the interventions, simulations were run 100 times with each posterior parameter sample and the mean outcome value obtained. As we used 2000 posterior samples from each parameter estimated by model fitting, this resulted in a total of 200,000 model runs for each simulated scenario.

## Acknowledgements

We thank the staff and patients' families at the Angkor Hospital for Children for their participation in the study. We also thank Anastasia Hernandez Koutoucheva and Patrick Musicha for their comments on the manuscript.

## Additional information

### Funding

| Funder | Grant reference number | Author |
| --- | --- | --- |
| Wellcome | 106698/Z/14/Z | Nicholas PJ Day |
| Medical Research Council | MR/K006924/1 | Ben S Cooper |

The funders had no role in study design, data collection and interpretation, or the decision to submit the work for publication.

### Author contributions

Thomas Crellen, Software, Formal analysis, Visualization, Methodology; Paul Turner, Conceptualization, Resources, Data curation, Supervision, Investigation; Sreymom Pol, Resources, Project administration; Stephen Baker, To Nguyen Thi Nguyen, Nicole Stoesser, Resources, Data curation; Nicholas PJ Day, Supervision, Funding acquisition; Claudia Turner, Conceptualization, Funding acquisition, Investigation, Project administration; Ben S Cooper, Conceptualization, Resources, Data curation, Supervision, Funding acquisition, Investigation, Methodology

### Author ORCIDs

Thomas Crellen (iD) https://orcid.org/0000-0003-2934-1063
Paul Turner (iD) http://orcid.org/0000-0002-1013-7815
Sreymom Pol (iD) https://orcid.org/0000-0001-8393-659X
Stephen Baker (iD) http://orcid.org/0000-0003-1308-5755
Ben S Cooper (iD) https://orcid.org/0000-0002-9445-7217

### Ethics

Human subjects: Written consent was obtained from mothers before study enrolment. The study was reviewed and approved by the Angkor Hospital for Children Institutional Review Board (1055/13 AHC) and the University of Oxford Tropical Ethics Committee (1047-13).

## Decision letter and Author response

Decision letter https://doi.org/10.7554/eLife.50468.sa1
Author response https://doi.org/10.7554/eLife.50468.sa2

---

# Additional files

### Supplementary files

• Transparent reporting form

### Data availability

Code for reproducing the statistical model fitting and anonymised patient data are available at https://github.com/tc13/transmission-dynamics-klebsiella (copy archived at https://github.com/eli-fesciences-publications/transmission-dynamics-klebsiella). The code for the agent based model and parameter values for forward simulations are available at https://github.com/tc13/ward-infection-ABM (copy archived at https://github.com/elifesciences-publications/ward-infection-ABM). Short read sequence data is available from NCBI under accession numbers PRJNA395864 and PR600JEB24970. Genomic data can be visualised in the Microreact project https://microreact.org/project/BV05TIXkU.

The following dataset was generated:

| Author(s) | Year | Dataset title | Dataset URL | Database and Identifier |
|---|---|---|---|---|
| Turner P, Baker S, Cooper B | 2018 | Whole genome sequencing of Klebsiella pneumoniae collected from patients and environment in a hospital in Cambodia | https://www.ncbi.nlm.nih.gov/bioproject/PRJEB24970 | NCBI BioProject, PRJEB24970 |

The following previously published datasets were used:

| Author(s) | Year | Dataset title | Dataset URL | Database and Identifier |
|---|---|---|---|---|
| Turner P, Stoesser N, Cooper B | 2016 | Klebsiella pneumoniae on a Cambodian neonatal unit | https://www.ncbi.nlm.nih.gov/bioproject/?term=PRJNA395864 | NCBI BioProject, PRJNA395864 |

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
