## [Decision Letter]

**Acceptance summary:**

This manuscript addresses the important question of transmission of drug resistance in a hospital setting. It analyses a dataset of children regularly swabbed for *Klebsiella* pneumoniae after admission to an intensive care unit in Cambodia. Such data is inherently complex and difficult to interpret. The authors use mathematical modelling approach to investigate the role of a number of factors such as colonization pressure, antibiotic treatment, and breastfeeding. The results provide insights into the spread of drug resistant bacteria in an important setting for the development and transmission of drug resistant bacteria.

**Decision letter after peer review:**

Thank you for submitting your article "Transmission dynamics and control of multidrug-resistant *Klebsiella* pneumoniae in neonates in a developing country" for consideration by *eLife*. Your article has been reviewed by four peer reviewers, including Miles P Davenport as the Reviewing Editor, and the evaluation has been overseen by a Neil Ferguson as the Senior Editor. The following individuals involved in review of your submission have agreed to reveal their identity: Jon Zelner; Edward J Feil.

The reviewers have discussed the reviews with one another and the Reviewing Editor has drafted this decision to help you prepare a revised submission.

Summary:

This manuscript uses a modelling approach to investigate acquisition of drug resistant bacteria in a hospital setting. The authors study various risk factors for 'acquisition', and find that antibiotic use was the only significant effect. The strength of this study is that it looks at a very clinically important organism in the exact setting in which it is problematic. The value of the study is tempered by the fact that much of the screening data was already published, along with more coarse-grained analysis of the role of antibiotic on acquisition. The modelling and analysis is therefore a central factor in the study, but also falls short in several ways that need to be addressed. The reviewers expressed that there are several possibilities to improve the analysis that need to be explored.

Essential revisions:

1) the Bayesian approach to model fitting of the risk factor and the transmission models, uses arbitrary priors, that do not truly reflect prior probability. Despite the authors claim that they were selected to be weakly informative, most prior distributions cover a range that make it possible that they are significantly influencing the posterior. A straightforward way of conveying this to the reader is to plot the prior and the posteriors together. This is a key shortcoming because if the model estimates are driven by these arbitrary choices, much of the remaining paper comes into question. This is certainly a criticism that can be addressed by changing the fitting approach to remove the influence of the priors, or by switching to another approach, maximum likelihood or proportional hazard models.

2) the results of the model are not presented carefully, many of the risk factors have credibility intervals that would not generally be considered significant, yet are presented as if they are (e.g.: "covariates associated with reduced daily risk of acquisition," but nothing in this paragraph is significant; "suggests that contamination left by previously colonized infants decays rapidly to background levels", when there is no support for the model with this parameter and the λ estimate is heavily influenced by the prior with mean 1 and variance 2). This is also true for the results presented in Figure 4B, C, D and the lengthy discussion in Discussion section, which fails to accurately convey the possibility that these factors play no role in colonization is also consistent with the data. The large uncertainty in most parameter estimates likely reflects the challenge of fitting 109 observations with a 14-parameter model. This needs to be stated and the limitations emphasized.

3) the analysis of the transmission models appears to have a bias toward finding model 4 to be the best because there is a single value of α for all the sequence types, but there are different average colonization probabilities across the sequence types (because some had many more observed colonization events than others). This variability it likely to be picked up, at least to some extent, in the ST specific betas. A fairer test is whether ST specific β fits better than a ST specific alphas. The temporal clustering of ST in Figure 3F suggest this may be true, but the approach seem to have been biased to find an effect of β. Are there explanations for the temporal clustering of the ST other than transmission in the NU, such a transient increased in the community or in the water supply etc.

4) A concern with this analysis relate to the way the different infection models are described. Specifically, the differentiation between a 'risk factor model' and a 'transmission model'. It is not clear from the main text that the transmission models considered are essentially hierarchical regression models in which the impact of an increasing number of colonized patients is considered to have an additive rather than a multiplicative effect on individual risk. This is not a critique of the modeling, which is generally appropriate and well-done. But it is difficult to read the manuscript and understand the distinction between these two sets of analyses. In addition, the authors should clarify earlier that the non-significant, negative OR for colonization pressure may be due to this fact as a way of framing the need for an additive model that more able to directly account for the impact of transmission on daily infection risk.

5) It is not clear to me why the transmission models could not incorporate some of the covariates used in the first set of risk factor models. For example, age at admission or the number of nurses on the unit could be used to modulate infection risk, i.e. be used to calculate an individual risk ratio that is then multiplied by the hazard of infection, e.g. exp(\ζ X_i_)*(\α +\β n_i_), where X_i_ is a set of individual or day-specific risk factors, and \ζ is a vector of log-risk ratios. This would seem to make the most of the data and incorporate both the additive effect of colonization pressure with the risk-modulation of individual level factors.

---

## [Author Response]

Summary:This manuscript uses a modelling approach to investigate acquisition of drug resistant bacteria in a hospital setting. The authors study various risk factors for 'acquisition', and find that antibiotic use was the only significant effect. The strength of this study is that it looks at a very clinically important organism in the exact setting in which it is problematic. The value of the study is tempered by the fact that much of the screening data was already published, along with more coarse-grained analysis of the role of antibiotic on acquisition. The modelling and analysis is therefore a central factor in the study, but also falls short in several ways that need to be addressed. The reviewers expressed that there are several possibilities to improve the analysis that need to be explored.

We thank the four reviewers, including the reviewing and senior editors for their comments and criticisms of our article. We have taken these points into consideration and conducted a number of additional analyses to confirm the robustness of our results to different modelling decisions (e.g. prior distributions). For other criticisms we consider it more appropriate to include these as study limitations in the Discussion.

While the individual points are dealt with below, we would like to respectfully challenge the reviewers’ reference to statistical significance, e.g. in the Summary and Essential revisions 2 and 4. The limitations and misuse of significance testing has been widely discussed by the statistical community, please see “American Statistical Association [ASA] statement on Statistical Significance and p-values”^1^, and discussions by Gelman^2,3^, McElreath^4,5^ and Spiegelhalter^6^. In line with current statistical thinking which recommends that statistical “bright lines” (such as p<0.05) are avoided, we make no reference to “significant” or “non-significant” results in our article. Rather we consider the full posterior distribution to be informative about parameter estimates, in addition to interpreting our results in the context of the existing literature. In this analysis we are not setting up null models to be refuted, but rather estimating bio-medically relevant parameters based on our data and the model structure. We do not consider any of our models to be correct, but instead provide a number of possible models and use an information criterion (and our background biological knowledge) as a guide for the most appropriate model. This is in line with best practices in ecological modelling that have been established for some time^7,8^. We hope therefore that the reviewers will consider our analyses and the response to their comments in this light.

Essential revisions:1) the Bayesian approach to model fitting of the risk factor and the transmission models, uses arbitrary priors, that do not truly reflect prior probability. Despite the authors claim that they were selected to be weakly informative, most prior distributions cover a range that make it possible that they are significantly influencing the posterior. A straightforward way of conveying this to the reader is to plot the prior and the posteriors together. This is a key shortcoming because if the model estimates are driven by these arbitrary choices, much of the remaining paper comes into question. This is certainly a criticism that can be addressed by changing the fitting approach to remove the influence of the priors, or by switching to another approach, maximum likelihood or proportional hazard models.

We agree that plotting the prior and posterior distributions together is a useful method to visualise the effect of the prior distribution on the result, and we have done this as suggested (see Figure 2—figure supplement 2 and Figure 3—figure supplement 2). Our priors were chosen on the basis of best practice guidelines provided by leading statisticians, including the creators of the STAN fitting software^9^. The most important validation of our model is the posterior predictive distribution (Figure 2D), which shows that the Risk Factor Model closely predicts the observed number of colonisation events.

While the choice of prior distributions remains an area of active debate, it is important to note that the prior will always influence the posterior. Prior distributions are an important tool for i) incorporating prior beliefs or information into the model, ii) preventing the model from exploring regions of parameter space that are biologically implausible, and iii) avoiding overfitting. Indeed, the use of weakly informative priors is directly analogous to the use of standard regularisation methods in frequentist analysis. Methods from frequentist analysis mentioned by the reviewers (maximum likelihood, proportional hazards models) are not somehow “more objective” and contain their own, although often less explicitly stated, assumptions. Using diffuse or uniform (“flat”) priors with a large range in Bayesian inference will typically give similar results to maximum likelihood. However, the use of “flat” or uniform priors is not generally recommended^10^.

We do, however, fully agree that there is value in exploring sensitivity of results to the choice of prior distributions. Following this criticism of the reviewers, we have therefore explored different prior distributions for Risk Factor Model A to test the sensitivity of the result to the prior. Varying the priors for Risk Factor Model A to normal distributions with much lower variance, as suggested, results in only small changes to the posterior distributions (in Figure 2—figure supplement 2); WAIC is also unaffected.

2) the results of the model are not presented carefully, many of the risk factors have credibility intervals that would not generally be considered significant, yet are presented as if they are (e.g.: "covariates associated with reduced daily risk of acquisition," but nothing in this paragraph is significant; "suggests that contamination left by previously colonized infants decays rapidly to background levels", when there is no support for the model with this parameter and the λ estimate is heavily influenced by the prior with mean 1 and variance 2). This is also true for the results presented in Figure 4B, C, D and the lengthy discussion in Discussion section, which fails to accurately convey the possibility that these factors play no role in colonization is also consistent with the data. The large uncertainty in most parameter estimates likely reflects the challenge of fitting 109 observations with a 14-parameter model. This needs to be stated and the limitations emphasized.

In line with current statistical thinking we think it is inappropriate to dichotomise results into significant/ non-significant (see comment above and statement from American Statistical Association^1^), and so consider our presentation and discussion of the posterior odds ratio distributions appropriate. We disagree that there is “no support” for transmission model 3; there is less support according to WAIC, but due to the interest in colonisation with resistant *K. pneumoniae* resulting from contact with contaminated surfaces in hospitals^11^, we consider it is appropriate to discuss the estimates from this model. The results from our analysis should not be interpreted from a frequentist perspective; posterior distributions inherently account for uncertainty in the parameter estimates. For instance if 6% of the posterior odds distribution for a covariate is <1, then there is a 94% probability that the covariate increases the risk of acquisition / detection. We have made this clearer in the text; in the Materials and methods subsection “Statistical Model Fitting” we add “Uncertainty in parameter estimates is represented by the posterior distributions, and the estimated probability that a particular parameter is within a certain range is given by the area under the curve of that parameter’s marginal posterior distribution within that range.”

To address the reviewers’ concerns that the estimate of λ in transmission model 3 is heavily influenced by the prior distribution, we have re-fit the model using a less informative prior (Figure 3—figure supplement 2. The model estimate of the *K. pneumoniae* half-life on surfaces changes from 3.6 hours with our initial prior distribution (normal(1,2)) to 2.5 hours with a less informative prior (normal(0,5)). Our interpretation of the results from these models is qualitatively the same (“*K. pneumoniae* surface contamination decays quickly to background levels”), and altering the prior distribution had no effect on WAIC. The only published value from experimental studies for resistant *K. pneumoniae* half-life on surfaces gave a value of ~12 hours^12^, which is equivalent to a parameter estimate of 1.39 in our model. Our initial prior distribution (normal(1,2)) therefore gives more weight to biologically plausible areas of parameter space, and we continue to use the result from this model in the main text. We have expanded the text to better explain our prior choice for this model– see Materials and methods subsection “Transmission Models Incorporating Sequence Type Data”.

3) the analysis of the transmission models appears to have a bias toward finding model 4 to be the best because there is a single value of α for all the sequence types, but there are different average colonization probabilities across the sequence types (because some had many more observed colonization events than others). This variability it likely to be picked up, at least to some extent, in the ST specific betas. A fairer test is whether ST specific β fits better than a ST specific alphas. The temporal clustering of ST in Figure 3F suggest this may be true, but the approach seem to have been biased to find an effect of β. Are there explanations for the temporal clustering of the ST other than transmission in the NU, such a transient increased in the community or in the water supply etc.

We agree that it would be interesting to explore variance in the α parameter and if this results in a better fit to data by WAIC than variance in β. We have therefore added a model which allows α to vary by sequence type (transmission model 5, described in Materials and methods). The WAIC from this model is much higher than from transmission models 2-4 (1793, compared with 1739, 1740 and 1733), indicating a poor fit to data (shown in Table 3).

We agree that other explanations for the temporal clustering of STs are possible, though less parsimonious. Overall the estimate of α (background contamination in the absence of colonised infants in the ward) is very low in all models, suggesting that effects other than person-to-person transmission are minimal in spreading resistant *K. pneumoniae* around the ward. Transmission from the community may be driving imported cases into the ward, and the variability of the *K. pneumoniae* population that we observe in the sequence data (Figure 3A) may reflect a large, diverse population of community *K. pneumoniae* that has not yet been investigated. We have added a sentence to the Discussion that “We cannot entirely rule out less parsimonious explanations for the temporal clustering of STs, such as a transient increase of certain STs in the water supply, though the person-to-person transmission route is most strongly supported by our models.”

4) A concern with this analysis relate to the way the different infection models are described. Specifically, the differentiation between a 'risk factor model' and a 'transmission model'. It is not clear from the main text that the transmission models considered are essentially hierarchical regression models in which the impact of an increasing number of colonized patients is considered to have an additive rather than a multiplicative effect on individual risk. This is not a critique of the modeling, which is generally appropriate and well-done. But it is difficult to read the manuscript and understand the distinction between these two sets of analyses. In addition, the authors should clarify earlier that the non-significant, negative OR for colonization pressure may be due to this fact as a way of framing the need for an additive model that more able to directly account for the impact of transmission on daily infection risk.

We agree that there is a need to clarify the analysis earlier in the manuscript and frame the reasoning for our additive transmission models. We have amended the text at the end of the Introduction to better introduce the different types of statistical models that follow in the analysis; “We fit four models with logit link functions to estimate the impact of covariates on the daily risk of acquisition or detection of 3GC-R *K. pneumoniae s.l.*. As these models are unable to identify the force of infection, and genomic data show the ward 3GC-R *K. pneumoniae s.l.* to be a highly heterogeneous bacterial community, we then fit five linear transmission models to estimate the force of infection for acquisition or detection of each ST.”

5) It is not clear to me why the transmission models could not incorporate some of the covariates used in the first set of risk factor models. For example, age at admission or the number of nurses on the unit could be used to modulate infection risk, i.e. be used to calculate an individual risk ratio that is then multiplied by the hazard of infection, e.g. exp(ζ X_i_)*(α +β n_i_), where X_i_ is a set of individual or day-specific risk factors, and ζ is a vector of log-risk ratios. This would seem to make the most of the data and incorporate both the additive effect of colonization pressure with the risk-modulation of individual level factors.

We did not include additional covariates when fitting the transmission models, as the *K. pneumoniae* sequence data was only taken within a four month window of the year-long study. In this period there were 171 first acquisition/detection events for different STs, however given the number of ST at-risk patient days was >48,000, the events were much sparser than the 109 acquisition events over 864 patient days in the risk factor model and are likely to have wider uncertainty intervals. As the underlying data are different in the risk factor and transmission models and as the models use different link-functions, the estimates of covariates will not be exactly the same and we considered that presenting two sets of odds ratios for the risk factors might confuse the reader.

We performed analysis similar to that described by the reviewer above to obtain the parameters for the individual-based model; we varied the force of infection inferred from the transmission models with the effect of covariates inferred from the risk factor models, by converting all parameters onto the log-odds scale to create a combined linear function and then using the inverse-logit function to obtain a probability of acquisition. This is described in the Materials and methods as “For simulations that explored the impact of interventions, the effect of each of the covariates of interest (probiotic consumption, breast milk feeding and number of nurses on the ward) were obtained from risk factor model A (Table 2 and Figure 2A). The marginal effects of each risk factor were transformed into a log-odds ratio and used to modify the daily risk of acquisition sampled from a β distribution with shape hyper-parameters α_s_ and β_s_.”

**References**

1) Ronald L. Wasserstein and Nicole A. Lazar (2016) The ASA Statement on p-Values: Context, Process, and Purpose, The American Statistician, 70:2, 129-133, DOI: 10.1080/00031305.2016.1154108

2) Blakeley B. McShane, David Gal, Andrew Gelman, Christian Robert and Jennifer

L. Tackett (2019) Abandon Statistical Significance, The American Statistician, 73:sup1, 235-245, DOI: 10.1080/00031305.2018.1527253

3) Gelman, A. and Stern, H., 2006. The difference between “significant” and “not significant” is not itself statistically significant. The American Statistician, 60(4), pp.328-331.

4) McElreath, R., 2018. Statistical rethinking: A Bayesian course with examples in R and Stan. Chapman and Hall/CRC.

5) Smaldino, P.E. and McElreath, R., 2016. The natural selection of bad science. Royal Society open science, 3(9), p.160384.

6) Matthews, R., Wasserstein, R. and Spiegelhalter, D., 2017. The ASA’s p-value statement, one year on. Significance, 14(2), pp.38-41

7) Hilborn, R. and Mangel, M., 1997. The ecological detective: confronting models with data. Princeton University Press

8) Bolker, B.M., 2008. Ecological models and data in R. Princeton University Press

9) https://github.com/stan-dev/stan/wiki/Prior-Choice-Recommendations

10) Gelman, A., Carlin, J.B., Stern, H.S., Dunson, D.B., Vehtari, A. and Rubin, D.B., 2013. Bayesian data analysis. Chapman and Hall/CRC.

11) Freeman, J.T., Nimmo, J., Gregory, E., Tiong, A., De Almeida, M., McAuliffe, G.N. and Roberts, S.A., 2014. Predictors of hospital surface contamination with Extended-spectrum β-lactamaseproducing *Escherichia coli* and Klebsiella pneumoniae: patient and organism factors. Antimicrobial resistance and infection control, 3(1), p.5

12) Weber, D.J., Rutala, W.A., Kanamori, H., Gergen, M.F. and Sickbert-Bennett, E.E., 2015. Carbapenem-resistant Enterobacteriaceae: frequency of hospital room contamination and survival on various inoculated surfaces. infection control and hospital epidemiology, 36(5), pp.590-593